Corrected: Publisher Correction

# Breaking the speed limit with multimode fast scanning of DNA by Endonuclease V

Arash Ahmadi [1], Ida Rosnes[1,2], Pernille Blicher[1], Robin Diekmann [3,4], Mark Schüttpelz [3], Kyrre Glette [5], Jim Tørresen [5], Magnar Bjørås[2,6], Bjørn Dalhus [1,2] & Alexander D. Rowe[1,7]

In order to preserve genomic stability, cells rely on various repair pathways for removing DNA damage. The mechanisms how enzymes scan DNA and recognize their target sites are incompletely understood. Here, by using high-localization precision microscopy along with 133 Hz high sampling rate, we have recorded EndoV and OGG1 interacting with 12-kbp elongated λ-DNA in an optical trap. EndoV switches between three distinct scanning modes, each with a clear range of activation energy barriers. These results concur with average diffusion rate and occupancy of states determined by a hidden Markov model, allowing us to infer that EndoV confinement occurs when the intercalating wedge motif is involved in rigorous probing of the DNA, while highly mobile EndoV may disengage from a strictly 1D helical diffusion mode and hop along the DNA. This makes EndoV the first example of a monomeric, single-conformation and single-binding-site protein demonstrating the ability to switch between three scanning modes.

[1] Department of Medical Biochemistry, Institute for Clinical Medicine, University of Oslo, NO-0372 Oslo, Norway. [2] Department of Microbiology, Oslo University Hospital HF, Rikshospitalet and University of Oslo, PO Box 4950 Nydalen, NO-0424 Oslo, Norway. [3] Biomolecular Photonics, Department of Physics, University of Bielefeld, Universitätsstraße 25, DE-33615 Bielefeld, Germany. [4] European Molecular Biology Laboratory (EMBL), Cell Biology and Biophysics Unit, Meyerhofstraße 1, DE-69117 Heidelberg, Germany. [5] Department of Informatics, University of Oslo, PO Box 1080 Blindern, NO-0316 Oslo, Norway. [6] Department of Clinical and Molecular Medicine, Faculty of Medicine and Health Sciences, Norwegian University of Science and Technology (NTNU), PO Box 8905, NO-7491 Trondheim, Norway. [7] Department of Newborn Screening, Division of Child and Adolescent Medicine, Oslo University Hospital, PO Box 4950 Nydalen, NO-0424 Oslo, Norway. Correspondence and requests for materials should be addressed to B.D. (email: bjornda@medisin.uio.no) or to A.D.R. (email: alerow@ous-hf.no)

Chemical alterations of DNA can introduce permanent damage and mutations, the accumulation of which may cause genomic instability—a primary cause of both cancer and aging[1-3]. In order to maintain genomic stability, DNA damage is corrected by several classes of DNA repair proteins[3]. Despite decades of intensive research, a detailed understanding of the molecular mechanisms by which these proteins identify and repair errors in DNA with incredible fidelity, while relying only on thermal energy to reach their targets, remains elusive. Facilitated diffusion is the widely accepted term used to describe the mechanism by which DNA binding proteins stochastically and passively locate target sites[4-6]. In this model the linear structure of DNA provides a track to which the proteins associate, enabling them to scan DNA with a dramatically increased rate of damage recognition compared to a random three-dimensional (3D) search[5,6]. According to an early theoretical outline of this model[4], proteins initially undergoing free 3D diffusion in solution, and bind to DNA at nonspecific sites. They then follow one of several strategies to scan DNA. (i) Proteins remain bound and scan the DNA driven by Brownian diffusion, following the DNA helix—known as one-dimensional (1D) helical sliding. Previous single-molecule studies have shown that helical sliding is the preferred strategy for several DNA repair proteins[7-9]. (ii) Proteins undergo repeated microscopic dissociation-reassociation steps during which they disengage from, but remain proximate to, the DNA with a high probability of reassociation at another site within the same region—this is referred to as hopping[10] or jumping[11]. (iii) Proteins transfer between two different segments of DNA due to close contact of those segments—known as intersegmental transfer[12]. (iv) Proteins macroscopically dissociate from DNA and continue with free 3D diffusion in solution. Despite some single-molecule studies showing that selected proteins choose either sliding or hopping for scanning[7-10], several bulk experiments[13,14] and theoretical[15-18] studies strongly support the idea that the most efficient target site recognition mechanism incorporates sliding interspersed with varying lengths or forms of hopping. Moreover, recent single-molecule studies have described proteins that can switch between hopping and sliding. However, this has so far only been attributed either to multimeric, ring-shaped proteins which encircle the DNA, keeping the unbound protein in close contact with DNA[11,19-21]; to proteins with two distinct binding sites[22]; or to proteins which undergo cofactor-induced conformational changes[12,23,24]. It remains to be determined whether interspersed 1D helical sliding and hopping is a strategy limited only to the classes of protein and complexes for which it has already been described, or whether it is also a strategy which may be adopted by single-conformation, monomeric proteins with a single DNA binding site.

In addition to nonspecifically associating with the DNA and performing rapid scanning, proteins must ideally be able to accurately interrogate the substrate to find specific target sites which they can stably bind. A central theoretical study addresses this issue by introducing two different modes of scanning, which they term "search" and "recognition" modes[17]. In the search mode, proteins interact with a smooth binding energy landscape of around $1\,k_{B}T$, over which they can easily perform efficient helical sliding. When the binding energy landscape is rougher ($>2\,k_{B}T$), scanning becomes extremely slow, and with increased roughness, the protein–DNA target complex tends towards a stable association–recognition mode. These two modes are originally described for two distinct protein–DNA binding conformations which may be alternated between on account of small or large conformational changes to either protein, DNA or both components. These conformational changes alter the roughness of the binding energy landscape and are coupled to the switch between specific and nonspecific protein–DNA interactions. Several examples exhibiting these modes have been examined in single-molecule scanning experiments[7-9,11,22,25] and the roughness of the energy landscape for helical sliding is consistently

determined to be between 0.6 and 1.78 $k_{B}T$. The ubiquity of these parameters across different classes of enzymes which scan DNA, and the role of specific structural elements in the choice of scanning mode, are therefore central to our understanding of these interactions.

One such structural element is seen in the crystal structures of endonuclease V (EndoV) in complex with damaged DNA, which have revealed a strand-separating wedge motif at the protein surface, consisting of the four residues: Pro79, Tyr80, Ile81 and Pro82 (PYIP)[26]. This highly conserved PYIP wedge is crucial for EndoV's ability to recognize helical distortions in DNA, since the DNA strands are split exactly at the weak point in DNA[27], and can therefore be reasonably expected to play a role in switching from search to recognition mode. EndoV recognizes and cleaves the DNA strand next to a wide variety of DNA damage, ranging from deaminated bases[28-34], apurinic/apyrimidinic (AP) sites[29,32], base mismatches[32,35], insertion–deletion (ID) loops, hairpins, flaps and pseudo-Y structures[36]. In addition, the enzyme has also been shown to bind to, but not cleave, a variety of branched DNA structures such as forks, three-way junctions and Holliday junctions[37].

In the present study the scanning mechanism of EndoV has been characterized at the single-molecule level, and compared with the well-studied baseline human 8-oxoguanine DNA glycosylase 1 (hOGG1), which represents a typical protein with pure helical sliding behavior[7]. Two aspects of the scanning mechanism are investigated here. (i) The role of EndoV's wedge motif in switching between search and recognition modes is examined by comparison of wild-type EndoV (wt-EndoV) and a wedge-deficient mutant EndoV (wm-EndoV). (ii) The ability of these proteins to switch between hopping and sliding modes, despite being monomeric proteins with a single conformation and single DNA binding site and none of the other properties common to enzymes which have previously displayed this behavior. We show that the intercalating wedge motif is involved in rigorous probing of the DNA, and that the proteins scan DNA at speeds clearly exceeding the limit for helical sliding, making EndoV the first example of a monomeric protein able to switch between a strictly 1D helical diffusion and hopping along the DNA independent of a conformational change in the protein or by binding of a cofactor.

## Results and discussion

**Protein–DNA interaction visualization.** To examine protein–DNA interactions at the single-molecule level, DNA tracks were produced by attaching a 12 kbp fragment of λ-DNA to a microscope coverglass at one end and a polystyrene bead held in an optical trap at the other end (Fig. 1), allowing the DNA to be elongated to ~95% of its theoretical length. Protein molecules were fluorescently labeled with ATTO 647N, and the emission signal was detected using a high light collection efficiency optical setup[38] combined with effective noise reduction strategies, including surface passivation and illumination of a thin layer of the sample[39]. Once DNA was localized and linearized (Fig. 1c, top) the labeled proteins were injected into the sample cell of a custom-made flow system and interactions between proteins and DNA were recorded with high temporal resolution (7.5–23.5 ms) (Fig. 1c, middle and Supplementary Movie 1). All signals in a frame were localized with a spatial precision of between 20 and 42 nm, and the trajectories—defined as the path followed by a protein during a single uninterrupted DNA binding event—of proteins moving along DNA were analyzed using custom analysis routines[40] (Fig. 1c, bottom). We took several important steps to remove potential sources of error. In the current study, DNA binding events take place in the absence of flow, DNA-intercalating dyes and large antibodies and quantum dots, which heavily affect the protein's diffusive properties. By making use of photo-stable dyes with high quantum yield along with the above-mentioned strategies to increase signal-to-noise ratio, we have been able to exploit the advantages of long-lived bright

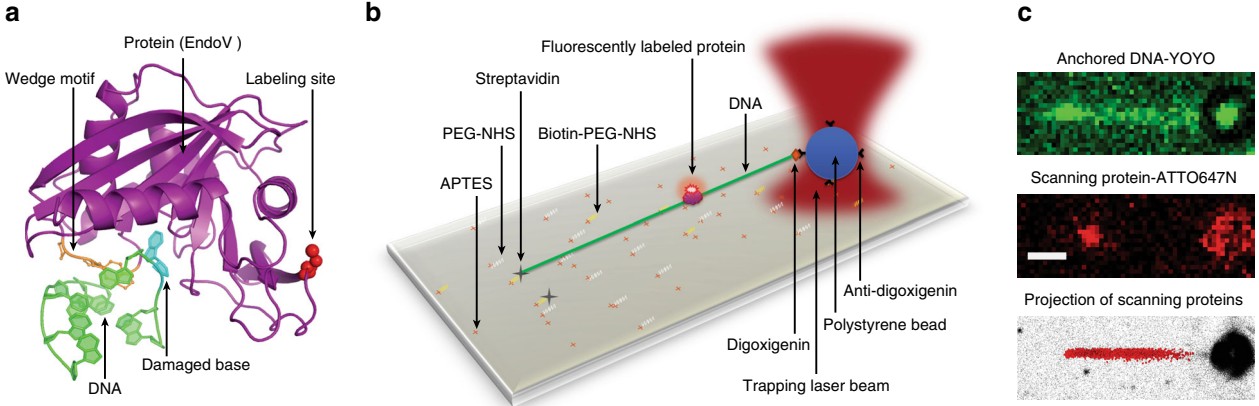

**Fig. 1** Protein–DNA interaction experiment. **a** Crystal structure of *Thermotoga maritima* (Tma) EndoV (purple) in complex with deaminated DNA (green). The protein labeling site (red spheres) does not interfere with DNA binding and damage recognition (cyan base in DNA). The PYIP-wedge motif (orange) consists of the four residues: Pro79, Tyr80, Ile81 and Pro82. **b** Single-molecule experimental setup. APTES (3-Aminopropyl)triethoxysilane, PEG-NHS *N*-hydroxylsuccinimide (NHS) functionalized polyethylene glycol, Biotin-PEG-NHS biotinylated PEG-NHS. The position of the polystyrene bead is controlled by a steerable optical trap, and used to linearize the DNA. **c** Upper panel: epifluorescence microscopy image of linearized DNA labeled with intercalating dye YOYO-1 (not present during the protein scanning experiments). Middle panel: a single fluorescently labeled EndoV molecule interacting with linearized DNA. Lower panel: projection of all detected trajectories of one data set of EndoV along the DNA. White scale bar equals 1 μm

signals without compromising the biological relevance of our results by drastically increasing the size of the diffusing units.

**Instantaneous diffusion rate distributions**. The proteins displayed heterogeneous scanning behavior, pausing intermittently between qualitatively fast and slow runs on the DNA (Fig. 2a, insets). Instantaneous diffusion rates were calculated with moving windows 5 steps in length for individual trajectories (Fig. 2a and Supplementary Fig. 1), and were compared against a single-mode simulated random walk trajectory of a particle with an average diffusion coefficient equal to that measured for the corresponding protein ($D_{\text{ave-hOGG1}} = 0.127 \pm 0.269\ \mu\text{m}^2\,\text{s}^{-1}$, $D_{\text{ave-wt-EndoV}} = 0.645 \pm 0.977\ \mu\text{m}^2\,\text{s}^{-1}$, $D_{\text{ave-wm-EndoV}} = 0.701 \pm 0.936\ \mu\text{m}^2\,\text{s}^{-1}$). The histogram in Fig. 2a shows the distribution of calculated instantaneous diffusion rates for the three proteins, while the solid line of the overlaid density plot shows the distribution of a single-mode simulated random walk. The apparent shape of the distribution of proteins shows the deviation from that which would be expected in the case of a monomodal random walk. The instantaneous diffusion coefficients for proteins span a range a whole order of magnitude larger than the corresponding simulated random walks, and there are clear indications of multimodality.

**Classification of scanning by activation energy**. To investigate the existence of these modes we began by considering the observed 1D diffusion in terms of the binding energy landscape. According to a pure hydrodynamic model[41,42], when proteins follow a helical path along DNA, there is an upper limit for the observed 1D diffusion rate, calculated as $0.89\ \mu\text{m}^2\,\text{s}^{-1}$ and $1.3\ \mu\text{m}^2\,\text{s}^{-1}$ for hOGG1 and EndoV, respectively (Supplementary Note 1). In this model, friction between the DNA and the protein is assumed to be zero—ideal sliding with no activation energy barrier. In reality, proteins experience an energy barrier landscape of the order of $k_B T$[17], yielding measured diffusion constants which are markedly lower than the theoretical upper limit. Relating this ideal upper limit to the measured diffusion rates gives the corresponding value of the instantaneous activation energy barrier. Considering the sliding process in terms of the kinetics of a sequence of interactions between the protein and adjacent DNA bases gives a rate constant ($k$) for the interaction of $k = \frac{1}{t} = 2D/\langle x^2 \rangle$, where $t$ is time ($s$), the mean square displacement $\langle x^2 \rangle$ is equal to 1 bp² and the diffusion constant $D$ is

expressed in terms of bp² s⁻¹. From the Arrhenius equation we know that $k = A e^{-E_a/k_B T}$, and for ideal sliding at the upper diffusion rate limit where $E_a = 0$, $A = k_{\text{ideal}}$. The activation energy barrier is obtained as $E_a = \ln\left(\frac{k_{\text{ideal}}}{k}\right) k_B T$. Using the instantaneous diffusion rate results we are able to calculate $E_a$ for every step in the trajectories, as shown in Fig. 2a. Based on the value of $E_a$ these steps are classified to belong to one of these three ranges representing different modes of diffusion (Supplementary Movie 2). (i) $0.5\,k_B T < E_a < 2k_B T$—based on several theoretical and experimental studies[7–9,11,17,22,25,42], where the diffusion process is expected to be dominated by helical sliding. (ii) $E_a > 2k_B T$—describes segments where proteins apparently paused locally on the DNA or where step sizes were below the spatial resolution limit of our instrumentation, and this mode is referred to as recognition[17] or interrogation mode[9]. (iii) $E_a < 0.5 k_B T$—in this mode the protein slides very close to the upper limit of the diffusion rate, an observation which has not been reported previously for helical sliding. In the case of EndoV, in around 50% of the time in this mode, the protein exceeds the upper limit for diffusion by helical sliding ($E_a < 0$). This is a clear indication that in this mode, the protein diffuses in a manner which is incompatible with the limitations of helical sliding, while remaining clearly associated with the 1D axis of the DNA.

**Classification of scanning by hidden Markov model**. To further validate the three-mode interpretation of our scanning results, avoiding kinetic assumptions related to the activation energy barrier classification, we used a recently published variational Bayes single-particle tracking (vbSPT) method[43] to identify and classify diffusion states using a hidden Markov model (HMM) approach. The model assumes a given number of diffusion states with memoryless transitions occurring between them. Each state represents an independent Gaussian distributed component of the overall diffusion spectrum. By finding the best fit to the experimental data, the ideal number of states, their associated diffusion coefficients, occupancies and transition probabilities are determined using a variational Bayesian method. The vbSPT software determines the ideal number of states by over-specifying the initial number of hidden states, and converging on the most suitable value. While the vbSPT reliably selected a single mode as the best fit for our simulated random walks, the software failed to converge on a reasonable number of states for our experimental

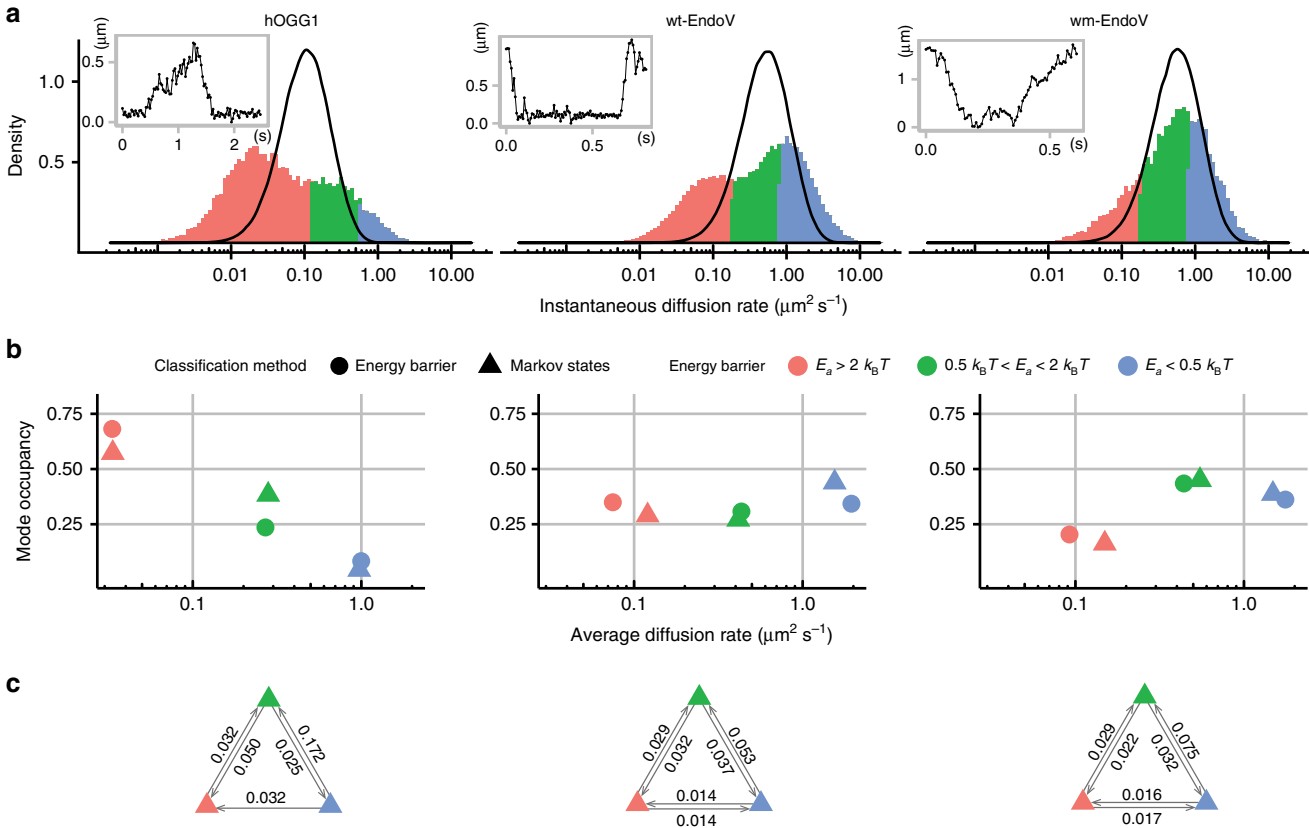

**Fig. 2** Diffusion analysis. **a** Distribution of instantaneous diffusion coefficients computed for hOGG1 (697 trajectories) with sliding window of 117.5 ms, and for wt- and wm-EndoV (wt-EndoV: 3443 trajectories, and wm-EndoV: 1099 trajectories) using sliding window of 37.5 ms. Insets: examples of single trajectories for each protein with time on the *x*-axis and displacement on the *y*-axis. **b** Comparison of classifications based on activation energy barrier (circles) and on Markov hidden state models (triangles), showing strong agreement between methods. The state occupancy is the proportion of time that proteins spend in each of the classified modes. The standard error of the mean of the average diffusion rate for all points is below 0.02 µm² s⁻¹. **c** Inter-state transition probabilities of the hidden Markov models. Arrows between triangles show transition probabilities for the transitions between the respective scanning modes, colored according to the energy barrier

data, and always selected more than three states to be the best fit for all proteins (8 states for wm- and wt-EndoV, and 6 states for hOGG1). We investigated the dynamics of mode switching for the three-state HMM classification. Looking at all possible transitions for wt- and wm-EndoV between three consecutive states, we find that while the transition matrix indicates roughly equal probabilities for molecules in state 2 (helical sliding) to switch into state 1 or state 3, the relative probability of switching back to the same state from which they entered state 2 is double the relative probability of switching into the third state (Supplementary Fig. 2). Thus, transitions from state 2 are dependent upon the transition into state 2 which may resemble some level of molecular memory. This manifestation of non-Markovian transitions could explain why vbSPT fails to converge on a reasonable number of states, instead adding states to approximate the non-Markovian behavior.

To further investigate the origin of excess states in vbSPT, we looked into the transition probability matrices for all HMM with 4 or more states to investigate the possibility of population heterogeneity. For a total of 13 models, all but 2 represented reducible transition matrices, while the last two models were nearly reducible (4-state models for wt- and wm-EndoV). In contrast to this, the 3-state models for all proteins investigated had irreducible transition matrices. Interestingly, we also find that vbSPT clearly prefers a 3-state hidden Markov model to a 2-state model, which resonates well with the energy barrier-based classification we employ here.

The results of both the energy barrier and the HMM classifications are shown in Fig. 2b. The average diffusion rate of each mode in the energy barrier-based classification is calculated by averaging the instantaneous diffusion rates over all steps within that particular mode. Since we use a moving window of five frames to calculate the instantaneous diffusion rate, the last four frames of each trajectory are not assigned any value for the instantaneous diffusion rate, and are thus excluded from the energy barrier classification. Because of this, we excluded the same four frames of each trajectory after the HMM classification, and recalculated the diffusion rate and occupancy of each Markov state. The recalculated values are very close to the original values reported by the HMM without frame correction (Supplementary Fig. 3), and the average diffusion rate and occupancy of each of the three Markov states are in good agreement with the initial energy barrier-based classification (Fig. 2b).

This consistency extends to the classification of individual points in trajectories, with 73% classification accuracy and a strong correlation between the results of the two independently performed classifications. The transition probabilities between the hidden Markov states for the three-state classifications are shown in Fig. 2c. For all three proteins the probability of switching from state 2 (helical sliding) to state 1 (interrogation mode) is consistently larger than for switching from state 3 (high mobility hopping) to interrogation mode. We also find that the transition

from high mobility hopping to helical sliding is more likely than going directly from hopping to interrogation mode. We interpret this as a clear indication that a helical sliding mode is central to efficient and reliable base interrogation and damage recognition.

**Role of EndoV wedge motif in scanning.** Despite an ideal expectation that the amplified fragments of DNA used in our experiments were undamaged, proteins clearly entered intermittent periods of confinement at randomly distributed locations along the DNA. These periods are represented by the $E_a > 2\,k_B T$, or interrogation (recognition) mode of scanning. As is shown in Fig. 2a, a large proportion of the instantaneous diffusion rate distribution for both wt-EndoV and hOGG1 lies within the interrogation mode, and a separate peak is clearly visible, around one order of magnitude higher. There is a clear distinction between the instantaneous diffusion rate distributions for these two proteins and the distributions of their corresponding single-mode simulated random walks. In contrast, the distribution for the wedge-deficient mutant EndoV (wm-EndoV) more closely resembles the corresponding simulated random walk than the others. The distribution for wm-EndoV lacks the clear interrogation peak, and this is reflected in the much lower interrogation-state occupancy calculated by both classification methods in Fig. 2b. In addition, using the classification of points along each trajectory from the hidden Markov-based classification, we can estimate the frequency of switching to the interrogation mode per 1000 bp DNA traversed. This is measured to be 1.51, 0.68 and 0.27 for hOGG1, wt-EndoV and wm-EndoV, respectively. These clear disparities between the scanning modes of wt- and wm-EndoV strongly suggest that the wedge motif plays a fundamental role in switching between search and interrogation mode. The crystal structure of wt-EndoV in complex with both deaminated DNA[26] and an insertion–deletion mismatch loop[27] show that the wedge penetrates the DNA helix to partly separate the two DNA strands at the site of lesion. The wedge locks the protein at the site of damage as it binds to an inherent weak point in the DNA. This is in line with the observation that EndoV might also recognize and process a large variety of DNA structures with anomalous DNA stacking such as AP-sites[29,32], base mismatches[32,35], ID loops, hairpins, flaps and pseudo-Y structures.[36] The high frequency of switching to the interrogation mode and the high occupancy of this mode for hOGG1, as calculated by both classification methods (Fig. 2b), corresponds well with the hypothesis that, in the interrogation mode, hOGG1 examines flipped-out bases one by one in an exosite pocket on the protein surface[44,45].

**Salt dependence of diffusion rate and hopping.** It has been suggested that friction between proteins and the DNA tracks on which they move typically reduces the diffusion rate by a factor of 2 to 5 compared to the theoretically calculated upper limit of diffusion for helical sliding[42]; this corresponds to an activation energy barrier of around $0.7-1.6\,k_B T$. The lowest experimental activation energy barrier reported for helical sliding has been 0.6 $k_B T$[22]; therefore we assume that an activation energy barrier of $0.5\,k_B T$ is a reasonable threshold, below which proteins cannot purely follow a helical path. In our experiments, EndoV and hOGG1 spend around 35% and 8% of the time in $E_a < 0.5\,k_B T$ respectively, of which ~50% belongs to a range where $E_a < 0$. This is clear evidence that for a considerable proportion of its interaction lifetime, EndoV is able to travel faster along DNA than would be possible if strictly limited to helical sliding. We hypothesize that in this mode, the protein is disengaged but remains electrostatically confined to the immediate vicinity of the DNA, enabling a striking increase in mobility through 1D diffusion utilizing the mechanism of hopping[4] (Fig. 3 and Fig. 4). Since higher ionic strengths increase the chance of dissociation, and reduce the chance of reassociation between DNA and protein, the occupancy of any diffusion state which depends on dissociation of the protein from the DNA should be elevated by higher salt concentrations. To verify whether significant hopping exists for the proteins under study, we investigated the salt dependence of the average diffusion rates for each of the three scanning modes classified according to the activation energy barrier. The average diffusion constant of EndoVs for scanning mode with $E_a < 0.5\,k_B T$ shows a 1.5-fold increase at higher salt concentrations (Fig. 3), while all other modes of diffusion for EndoV were unaffected. None of the diffusion modes for hOGG1 were salt dependent. This is strong evidence that EndoV utilizes hopping during the $E_a < 0.5\,k_B T$ mode, since a higher salt concentration

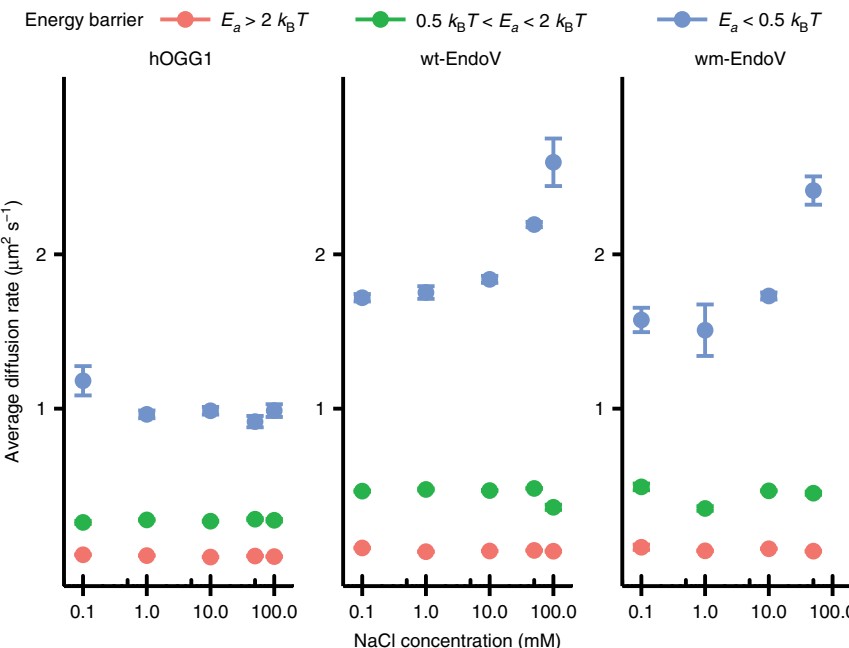

**Fig. 3** Diffusion rate analysis by buffer salt concentration. Average diffusion rates for each mode of diffusion as a function of buffer salt concentration. Error bars show the standard error of the mean (SEM) and are not visible when smaller than the size of the points in the plot

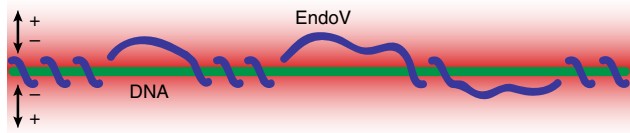

**Fig. 4** Model of interspersed helical sliding and hopping on DNA. The protein trajectory (blue trace) is confined within the radial electrostatic field (red to white) surrounding the DNA (green) to achieve rapid reassociation to DNA and efficient target localization during DNA scanning. Number of turns and length of hopping steps are unknown parameters and not to scale in this scheme

increases the amount of time spent in a higher mobility state, thereby increasing the average diffusion rate. On the other hand, low occupancy of the $E_a < 0.5\ k_BT$ state for hOGG1 indicates that helical sliding is the dominant scanning strategy for hOGG1, which is consistent with previous reports[7]. If hopping does occur for hOGG1, it is very short-lived and salt dependence cannot be detected within our experimental and analytical resolution.

**Multimode fast scanning and mode switching**. We have shown that the three modes of scanning based on activation energy barrier $E_a > 2\ k_BT$, $0.5\ k_BT < E_a < 2\ k_BT$, $E_a < 0.5\ k_BT$ are representative of DNA interrogation, helical sliding, and hopping, respectively. The wedge motif of EndoV plays an important role in switching from scanning to interrogation mode, and the protein uses a mix of helical sliding and short helically decoupled hopping in the fast DNA search mode; this suggests that the interactions between protein and DNA alternates between states with markedly different activation energy barriers, allowing distinct diffusion modes. Crystallographic data show that the free and DNA-bound EndoV are structurally similar[26], which rules out structural changes of the protein as the mechanism for switching between 1D hopping and helical sliding during scanning. Moreover, EndoV is a monomeric, single-binding-site protein too small to clamp around DNA, or delegate hopping and sliding to alternate binding sites or different conformational forms of the protein regulated by binding of a cofactor such as adenosine triphosphate (ATP), as seen for other proteins that are able to switch between helical sliding and hopping[11,12,19,20,22–24]. Despite this, the experimental evidence consistently shows interspersed phases of helical sliding and rapid 1D hopping for EndoV. This is compatible with a model for scanning which allows microscopic dissociation–reassociation events during which the protein remains constrained within a radial electrostatic field surrounding the DNA. Diffusing in 1D, but decoupled from the constraints of the helical path, the protein is able to adopt an even faster means of scanning the DNA (Fig. 4 and Supplementary Movie 2). These results have shown that there is significant detail in these protein–DNA interactions which has not previously been revealed due to technical limitations, and it will be extremely instructive to determine how ubiquitous these modes of interaction are among other classes of DNA-associated proteins.

## Methods
**Protein expression and purification**. *Escherichia coli* BL21 (DE3) RIPL Codon Plus cells (Stratagene) containing pET28b plasmids with the full-length sequences of *Thermotoga maritima* EndoV mutant D43A (wt-EndoV) or D43A/wedge mutant ($^{79}$PYIP$^{82}$→GGGG; wm-EndoV) were incubated in LB medium supplemented with 50 μg ml$^{-1}$ kanamycin at 37 °C. Protein expression was induced by adding 0.5 mM IPTG when the OD$_{600nm}$ reached ~0.75, followed by overnight incubation at 18 °C. Cells were harvested by centrifugation at 27,000 × *g*/4 °C for 20 min and resuspended in buffer A (50 mM NaCl, 20 mM MES, pH 6.5). Cells were disrupted by sonication and centrifuged at 27,000 × *g*/4 °C for 30 min. The protein extracts were incubated at 75 °C for 15 min in order to precipitate a large fraction of *E. coli* proteins, followed by a second centrifugation step. The supernatants were loaded onto a 5 ml HiTrap SP XL column (GE Healthcare) equilibrated with buffer A. The protein was eluted using a linear salt gradient to 1 M NaCl in buffer A. Fractions rich in *Tma* EndoV were pooled, concentrated and applied onto a Superdex75 size exclusion column (GE Healthcare) equilibrated

with buffer B (100 mM NaCl, 20 mM MES, pH 6.5). Protein fractions containing pure *Tma* EndoV were identified by sodium dodecyl sulfate–polyacrylamide gel electrophoresis (SDS-PAGE) gel.

The wedge mutant ($^{79}$PYIP$^{82}$→GGGG) was generated using the Quik-Change Site Directed Mutagenesis kit (Stratagene) and performed according to the manufacturer's protocol. The four glycine residues were simultaneously introduced into the full-length D43A EndoV sequence by using a complementary primer pair containing the glycine-specific codons (underlined) in the wedge position: 5′-AGGGGGAGAGATAACTTTTGGCGGCGGTGGGGGGCTCCTTGCTTTCAGA GAAGG-3′ and 5′-CCTTCTCTGAAAGCAAGGAGCCCCCCACCGC CGCCAAAAGTTATCTCTCCCCT-3′ (Eurofins MWG/Operon). The mutant construct was verified by sequencing, and the expression plasmid was subsequently transformed into *E. coli* BL21 (DE3) RIPL Codon Plus cells for protein expression. The inactive D43A mutant was used to avoid any disruptive cleavage of DNA substrates during scanning experiments.

An S118C mutant of a truncated 12-327 hOGG1 protein was generated using the Quik-Change Site Directed Mutagenesis kit (Stratagene) according to the manufacturer's protocol. The S118 was mutated to a Cys residue to allow attachment of the fluorescent ATTO N647 reporter dye. The primers used to produce the mutation were as follows: 5′-CACTGGG GTTCCGTGGACTGCCACTTCCAAGAGGTG-3′ and 5′-CACCTCT TGGAAGTGGCAGTCCACGGAACCCCAGTG-3′. The mutant construct was verified by sequencing and transformed into *E. coli* BL21 (DE3) RIL Codon Plus cells. The protein was purified as previously described[46].

**Fluorescent labeling of the proteins**. To directly observe the position of single proteins in real time, the fluorescent dye ATTO 647N (ATTO-TEC) was bound by maleimide-NH coupling to a native cysteine in the EndoV (C154), and a genetically engineered cysteine in Ogg1 (C118). The cysteines in both EndoV and hOGG1 are located on the protein surfaces far from the damage recognition pockets (active sites). Prior to labeling, all proteins were extensively dialyzed against 1× phosphate-buffered saline (PBS) buffer for 12–18 h at 4 °C. For the labeling procedure, 1 ml of 40–100 μM protein in 1× PBS buffer was mixed with 1.3-fold molar excess of ATTO 647N and incubated at room temperature in the dark for between 0.5 and 3 h. Free dye molecules were separated from the labeled proteins using a NAP-5 column (GE Healthcare) with 1× PBS as elution buffer. Fractions rich in labeled proteins were identified by measuring the absorbance at 280 and 647 nm (A$_{280}$ and A$_{647}$) using a NanoDrop One instrument (Thermo Scientific). The labeling efficiency for different batches of proteins varied between 30 and 80% as determined using A$_{280}$, A$_{647}$ and the molar extinction coefficients of the protein and the dye, respectively.

**λ-DNA substrate preparation**. In order to anchor one end of the DNA to the surface of a coverslip and attach the other end to a polystyrene bead, as shown in Fig. 1b, a 12 kbp linear fragment of λ-DNA with biotin and digoxigenin tags at each end was designed. This DNA substrate was prepared by PCR amplification of λ-DNA using primers modified with 5′ biotin (5′-bio-ACTTCGCCTTCTTCC CATTT-3′) and 5′ digoxigenin (5′-dig-ATCTCGCTTTCCACTCCAGA-3′) (Eurofins MWG/Operon). The PCR reaction was performed in a 1× LongAmp Taq Reaction Buffer, with a final volume of 50 μl, 300 μM of each dNTP, 0.4 μM of each primer, 2 units LongAmp Taq DNA polymerase (New England Biolabs) and 0.1 ng μl$^{-1}$ of λ-DNA template. The PCR included an initial denaturation step at 94 °C for 3 min, 35 cycles of denaturation (94 °C for 15 s), annealing (60 °C for 60 s) and primer extension (65 °C for 16 min), followed by a final extension step at 65 °C for 10 min. The quantity and quality of the PCR product was analyzed by spectrophotometry (260/280 absorbance measurements) and gel electrophoresis. The PCR product was stored in aliquots at 4 °C and diluted immediately before use.

**Polystyrene bead functionalization**. In order to attach a polystyrene bead to the digoxigenin-modified end of the λ-DNA, the beads were coated with anti-digoxigenin antibody. The method is mainly adapted from the manufacturers' manuals for the various materials used, in addition to Bangs laboratories' protocol for covalent coupling. Next, 50 μl of 100 mg ml$^{-1}$ carboxylate-modified polystyrene beads (diameter = 0.9 μm, Sigma-Aldrich) was suspended in 500 μl of MES buffer (50 mM MES, pH 6), mixed well and centrifuged at 10 × *g* for 5 min before the supernatant was discarded from the tube. This washing step was repeated three times. To make the surface of the carboxylate-modified beads amino-reactive, the beads were resuspended and incubated with 100 μl of 50 mg ml$^{-1}$ EDC (1-ethyl-3-(3-dimethylaminopropyl) carbodiimide hydrochloride, ThermoFisher) in MES buffer for 30 min at room temperature. To quench the reaction and remove the excess EDC, the beads were washed twice with MES buffer as explained above. Next, to facilitate the reaction of amino-reactive beads with the amine-containing antibody, the pH was adjusted to 8.5 by washing twice with borate buffer (0.1 mM sodium tetraborate, pH 8.5 adjusted with HCl). Then, the beads were resuspended and incubated with 100 μl of 0.5 mg ml$^{-1}$ anti-digoxigenin (Fab fragments from sheep, Roche) in borate buffer for 4 h with mild shaking at room temperature. After incubation, the beads were washed once with borate buffer, then washed and incubated with 50 mM TRIS at pH 8 for 2 h at room temperature. Finally, the beads were washed once and stored in 1× PBS (pH 7.5), 2 mg ml$^{-1}$ bovine serum

albumin (BSA) and 0.1% Tween-20. Batches with functionalized beads were stored at 4 °C and could be used for up to 1 month without any significant degradation of the functionality.

**Surface preparation for DNA anchoring**. In order to prepare coverslip surfaces for the anchoring of biotin-tagged DNA (Fig. 1b), and to passivate the surface preventing nonspecific binding, we performed a series of surface treatment operations including cleaning, functionalization and passivation. The method is adapted from previously published protocols[47–51] and further modified after extensive experimentation, and with input from the manufacturers' manuals.

(I) Initially, 6–8 coverslips (24 × 60 mm, Menzel Gläser) were placed in a glass-staining dish and rigorously cleaned in the following sequence: (i) sonication in pure ethanol for 30 min; (ii) rinsing with MiliQ filtered (MQF) water 3 times; (iii) sonication in 1 M potassium hydroxide for 30 min; and (iv) rinsing with MQF water 3 times. This series of cleaning steps (i)–(iv) were repeated 3 times over. Afterwards, the coverslips were carefully rinsed with acetone to remove any traces of previous solutions.

(II) To functionalize the surface with amine groups, the cleaned coverslips were incubated with 2% (v/v) solution of 3-aminopropyltriethoxysilane (Sigma-Aldrich) in dry acetone at room temperature for 2–4 min, immediately followed by thorough rinsing with MQF water to remove any residual acetone, then dried in a 100 °C oven for 30 min.

(III) Simultaneous functionalization for DNA binding and passivation towards nonspecific binding was achieved by saturating the coverslip surface with a mixture of amino-reactive polyethylene glycol (PEG-NHS, molecular weight (MW) = 5000 Da, Nanocs) and biotinylated PEG-NHS (Biotin-PEG-NHS, MW = 5000 Da, Nanocs). Then, 120 μl of a mixture of 150 mg ml$^{-1}$ PEG-NHS and 0.1 mg ml$^{-1}$ Biotin-PEG-NHS in 0.1 M sodium bicarbonate solution was sandwiched between two amino-functionalized coverslips and kept in a humid chamber overnight. The following day, the two coverslips were separated and washed carefully with MQF water and blow-dried with nitrogen gas. These PEGylated surfaces were stored in a vacuum and could be used for up to 3 weeks after preparation.

(IV) Prior to the main single-molecule experiment, 120 μl of streptavidin (from *Streptomyces avidinii*, Sigma-Aldrich) with a concentration of 0.01 mg ml$^{-1}$ in 1× PBS was added to the PEGylated surface of one coverslip, and sandwiched with another clean but not functionalized coverslip for 3–15 min, depending on the required density of binding sites. The PEGylated coverslip was washed, blow-dried with nitrogen and immediately used in the construction of the flow chamber.

**Flow chamber and surface-DNA-bead construction**. To construct the experimental flow chamber with a sample volume of around 20–30 μl, a streptavidin-coated coverslip was fused to a pre-cleaned microscope slide containing two holes for inlet and outlet flow on its surface, using double-sided tape. The chamber was connected to a 100 μl syringe (Model 1710 LT SYR, Hamilton) using silicon and PEEK tubes (1/16-inch outer diameter, Sigma-Aldrich). The syringe was connected to a pump (Harvard Apparatus PHD 2000) with sensitive control over the flow rate. A three-way valve (V100T, Upchurch Scientific) was positioned in the flow circuit allowing the user to switch the flow between a supply tank of solution and the reaction chamber. After assembly, the flow chamber was sequentially washed by flowing 200 μl of washing buffer (25 mM TRIS, pH 7.5) followed by 200 μl of blocking buffer (25 mM Tris-HCl, pH 7.5, 2 mM EDTA, 1–3 mg ml$^{-1}$ BSA, 0.01% (v/v) Tween-20) with a flow rate of 50 μl min$^{-1}$. The flow chamber was incubated with blocking buffer for 1–2 h. Following this, 200 μl of 5–50 ng ml$^{-1}$ of λ-DNA substrate was injected with a flow rate of 10 μl min$^{-1}$, and incubated for 5–30 min depending on the required density of DNA on the surface. Unbound DNA was washed away by the injection of 400 μl of 10% blocking buffer in washing buffer with a flow rate of 10 μl min$^{-1}$. Anti-digoxigenin-coated polystyrene beads with a concentration of 10–100 μg ml$^{-1}$ in 10% blocking buffer in washing buffer were mixed and sonicated for 30 s and injected with a flow rate of 5 μl min$^{-1}$. Depending on the overall efficiency of the sample preparations, beads bound to the anchored DNA molecules within 30–90 min. It is worth mentioning that high flow rates (more than 50 μl min$^{-1}$) were avoided to prevent ruptures of the surface-DNA-bead construction. In addition, sudden changes in the flow rate and air bubbles in the system were avoided to prevent anchored DNAs from being torn off the surface.

Excess unattached beads were carefully washed off with 200 μl washing buffer (containing the appropriate concentration of NaCl, depending on the experiment) with a flow rate of 5–20 μl min$^{-1}$. Finally, the fluorescently labeled proteins were added to the flow chamber with a concentration of 0.1–2 nM, in the appropriate assay buffer at a rate of 5–20 μl min$^{-1}$, and the interaction with the trapped DNA was recorded as described below.

High-concentration batches of labeled proteins (1–100 μM) were kept on ice, and immediately injected after dilution in assay buffer and subsequent thermal equilibration to room temperature. Note that care should be taken to avoid too much protein entering the system, since excess amount of protein in the flow chamber can lead to accumulation of protein molecules on the DNA and/or the coverslip surface, with an increase in the noise level and a drastically lower ability to detect and record single-molecule trajectories. It was also critical to constantly supply the reaction chamber with fresh protein.

**Holographic optical tweezers and imaging setup**. Individual flow chambers were fixed onto the piezo-steerable stage of a custom-built combined holographic optical tweezer and subdiffraction resolution microscope which has previously been described[38]. In brief, a spatial light modulator was used to steer the infrared trapping laser within the sample. The trapping beam and excitation laser were coupled into the microscope light path and the objective lens, allowing simultaneous trapping and fluorescence microscopy. The entry position of the beam into the objective lens was controlled using an adjustable mirror on the excitation light path, making it possible to generate a highly inclined and laminated optical sheet[39] for illumination of the sample. This resulted in concentration of laser power in the vicinity of the surface and a drastic drop in background noise from the sample volume, and was critical to achieving a high signal-to-noise ratio (SNR). The light emitted from the fluorophores was collected with the same objective lens and imaged on a high sensitivity EMCCD camera controlled by Micro-Manager software[52]. Taking advantage of high SNR along with high light collection efficiency detection, the exposure time of imaging was pushed down to 7.5 ms for EndoVs and 23.5 ms for hOGG1, allowing the observation and extraction of transient behaviors on very short times scales in the scanning process. With these exposure times, an average of 47–143, 42–132 and 45–173 photons were collected per frame for single wt-EndoV, wm-EndoV and hOGG1 molecules, respectively.

**Image processing and trajectory tracking**. Each data set (video stream recorded as consecutive TIFF files containing between 50,000 and 200,000 frames) was processed using the ThunderSTORM[53] plugin in FIJI[54]. All signals within each frame were localized by Gaussian fitting and registered according to the frame number (as a time series). The lower panel of Fig. 1c shows the projection of all signals in one data set into a single image. Localization precision (the average standard deviation of the peak of the Gaussian fits) of several immobile hOGG1, wt-EndoV and wm-EndoV were measured between 20–33 nm, 23–39 nm and 22–42 nm, respectively, to confirm localization precision.

Later, using code custom-developed[40] in R, the signal from diffusing proteins were tracked, separated from the background noise and registered as uninterrupted trajectories. For each data set (containing up to hundreds of trajectories occurring separately along a single DNA strand) in which the position of the bead and DNA remain unchanged, the beads were localized and several trajectories were visually inspected; these two details were then used to localize the DNA. Applying a rotation matrix, the coordinates were rotated such that the localized DNA was aligned along the x-axis, meaning that all trajectories along DNA were similarly oriented, and a boundary filter (200 nm either side of the DNA) was set on the y-axis to exclude molecules localized far from the target DNA. Within these limits, trajectories with signals that are present in more than 5 consecutive frames, and which had moved at least 300 nm along the DNA (at least once) during the trajectory, were retained and the remainder of analysis was conducted on this subset of the data. The effect of blinking of the fluorescent dyes was mitigated slightly by allowing two temporally adjacent trajectories to be connected if they were separated by only one frame, and the detected protein in these adjacent frames were localized within 600 nm of one another. Supplementary Movie 1 shows a sequence of trajectories which are cut from different data sets, concatenated and visualized using the TrackMate[55] plugin in FIJI.

**Instantaneous and average diffusion analysis**. In order to compute the instantaneous diffusion coefficient, a moving window of width 5 frames (37.5 ms for EndoVs and 117.5 ms for hOGG1) was used. As the window moves along the trajectories at each step the mean squared displacement ($<x^2>$) was calculated for the next 5 steps. From the diffusion equation[56] for 1D diffusion $<x^2> = 2Dt$, the diffusion coefficient was calculated as the instantaneous diffusion coefficient at that particular step. To compute the average diffusion coefficient of proteins in different diffusion modes the instantaneous diffusion coefficient was averaged over particular segments of trajectories belonging to either of modes.

**Code availability**. All source codes used in the analysis pipeline are available on GitHub: https://doi.org/10.5281/zenodo.1487773

## Data availability
The time series position data for all trajectories are available on Figshare: https://figshare.com/s/59402119aa83c4abb4c9. All data including the raw image data captured for this study are available from the corresponding authors upon reasonable request.

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

## Acknowledgements

We acknowledge South-East Norway Regional Health Authorities (grants 2014034 and 2015095 to B.D.), the Research Council of Norway (program FRIMEDBIO to M.B., program DAAD project 281255 to B.D.), the German Academic Exchange Service (grant no. 57402206 to M.S.) and the University of Oslo, MLS$^{UiO}$ program (PhD grant to A.A.) for funding. We thank Professor Oddmund Bakke at Oslo NorMIC imaging platform, University of Oslo, for access for initial testing of flow cells, DNA and proteins.

## Author contributions

A.A., M.B., B.D. and A.D.R. conceived the study; A.A., I.R., P.B. and B.D. prepared DNA and proteins; A.A., I.R., R.D., M.S. and A.D.R. contributed to single-molecule DNA scanning data collection; A.A., K.G., J.T., M.B., B.D. and A.D.R. participated in flow-cell design; A.A. and A.D.R. processed and analyzed the data; A.A., M.B., B.D. and A.D.R. interpreted the data, and discussed results and wrote the paper with contributions from the other authors.

## Additional information

**Competing interests:** The authors declare no competing interests. I.R. is currently employed by F. Hoffmann-La Roche (Roche Norge AS). The data included in this paper are based on research which has had no influence or involvement by F. Hoffmann-La Roche by any means. Any personal views of I.R. should not be understood or quoted as being made on behalf of or reflecting the position of F. Hoffmann-La Roche.

