## [Peer Review File · Nature Communications]

Reviewers' comments:

Reviewer #1 (Remarks to the Author):

The authors perform particle tracking experiments on enzymes involved in DNA repair. The main conclusion is that Endonuclease V has at least three distinct motion states associated with DNA: confined, slow, and fast. All three are passive and thermally driven. The fast state is speculated to be diffusive electrostatic gliding along DNA, which is faster than the standard helical sliding seen in related enzymes. The characteristics and dynamics of the motion states are significant as they directly influence the rate of repair; i.e., motion is optimized for searching out damaged segments of DNA. The main conclusion is supported by analysis of particle tracks obtained from fluorescence microscopy. The authors use fluorophores that have a negligible impact on the hydrodynamic radius of observed enzymes. The conclusion is first supported by computing "instantaneous diffusion" histograms over all particle tracks, which amounts to convolutionally estimating effective diffusion in a sliding window. Different window sizes were shown not to affect the qualitative shape of the histograms. The diffusion histograms for WT-EndoV was shown to be bimodal, with two well separated (~10 fold) peaks. A control *wm*-EndoV and a simulated control (Brownian motion) were shown to have a single peak. A similar double peak was observed for a related enzyme hOGG1. This data supports at least two modes of motion. No third mode is observed corresponding to fast motion. The justification for the third mode is the relative fraction of observed instantaneous diffusivities above the theoretical maximum for helical sliding (2% for hOGG1 and 14% for WT-EndoV).

To gather further justification for the three mode hypothesis, the authors use a machine learning method to classify motion states over 5 increment windows of a subset of tracks. A smaller subset of tracks is divided from the rest and hand segmented; each frame is labeled as confined, slow, or fast. Parameters of a random forest model are trained with the hand segmented tracks and then deployed on the remainder of the data. The total fraction of time the enzymes spend in each of the three states is computed from the classified tracks. The fraction of time in the confined state increases whereas the fast state is preserved between the *wt*-EndoV and the mutant, suggesting the wedge motif is essential for helical sliding but not fast motion.

Clearly, it is an important problem to understand dynamics of stochastic motion of DNA repair enzymes, and the authors have collected extremely valuable data. However, I am not convinced of their conclusion of a distinct third motion state. Unless substantial revisions are made, I cannot recommend publication.

1. The instantaneous diffusion result would better support a third motion state if there were three distinct peaks. The existence of 14% outliers above a theoretical maximum diffusivity is weak evidence given the many sources of noise that can affect tracking measurements. Minimally, one could show that a three state Markov process, with parameter fits from data, reconstitutes the observed histogram, while a best fit two state process does not.

I am aware the authors did experiments adding salt, but it is outside my expertise to say if this is significant evidence to support the main conclusion by itself, the tracking analysis certainly does not.

2. The machine learning approach (i.e., hand segmentation of tracks) inserts human bias into the classification scheme. Three motion states were classified because the model was trained by hand segmented tracks. At least the same bias is uniformly applied to all of the data. Sometimes this is the best that can be done. However, in this case, much better methods are readily available. Namely, the EM algorithm for the hidden Markov models approach, which is a systematic Bayesian formalism that has numerous advantages over hand segmentation: no hand segmentation is necessary, all of the assumptions are placed into a physical model (Markov process) of particle motion, and variational Bayes methods can even quantitatively select the best model (there is a

good Matlab package freely available from the Elf lab <http://vbspt.sourceforge.net/>). And I think these methods should be used to make the current work more objective and to encourage others to use the best methods available.

Reviewer #2 (Remarks to the Author):

The authors use single-molecule fluorescence imaging of a labeled DNA-binding protein (a bacterial endonuclease) on stretched DNA to visualize the molecular mechanisms with which the protein moves one-dimensionally along DNA. By an improved temporal resolution, they show that the protein transitions between two binding modes of 1d translocation: one that corresponds to helical scanning, the other to a nonhelical, microscopic hopping. While the observation is interesting and provides another example of how DNA-binding proteins have evolved to combine several search mechanisms to achieve overall kinetic efficiency in target identification, the work is largely confirming previously observed behavior of other systems.

In particular, similar work on the PCNA sliding clamp (Ref 17) demonstrated an alternation between helical and non-helical scanning, and other work on a DNA-bound phage polymerase (PMID: 20080681) showed that a similar transition could be induced by addition of a co-factor. The authors argue in their introduction that the behavior they observe with EndoV is different: Lines 57-59, "this has so far only been attributed either to...or to conformational changes" an argument repeated in their conclusion (lines 160-161). This reasoning is incorrect. Whatever the transitioning between the helical scanning mode and the 1d hopping mechanism, it is caused by a conformational change. It might be changing the tilt angle of a protein on the DNA (PCNA), binding/dissociation of a protein factor (the phage polymerase) or insertion/removal of a wedge motive (this work), it is still a conformational change. It might be thermally driven (line 63), it is still a conformational change...

The authors also ignore a hallmark paper in this field that provided more than a decade ago the theoretical rationalization for this thermally induced transition between scanning while sliding and not scanning while hopping (PMID: 15465864). Later experimental work (including the examples above) was interpreted in the context of this thermally-induced two-mode sliding model and has resulted in a picture of target identification on DNA that is now commonly accepted in the field. While interesting in its high time resolution and the direct observation of these two distinct diffusional behaviors, the work does not provide a significant advance in our understanding of the underlying mechanisms. The experiments are sound and the interpretation supported by the data. As such, my recommendation would to consider a more specialized journal.

- Line 43-44, "with a 100-fold increased rate". The authors should be careful with absolute statements such as this one. While 1d scanning can speed up target association, the rate with which this occurs is entirely dependent on the exact parameters, in particular the balancing between the association equilibrium constant of the protein with the DNA and the scanning rate constant, and the division between helical scanning and 1d hopping. A very tight association with the DNA while 1d scanning might even result in slower searches than exclusively 3d! See for example PMID: 15465864. This latter paper also lays the theoretical basis for the key observation in the current manuscript: the conformational transitioning between a mode that corresponds with sequence sampling and a mode that corresponds to mere 1d movement (without scanning). The authors should discuss this paper and place their results in the context of this theoretical framework.

- Lines 89-91, "the payload of large antibodies and quantum dots...used...in previous studies...have been replaced...". This is quite a disingenuous statement. A large part of the reported work in this area has been done using small-molecule organic dye molecules.

- Line 120, "Around 1% of the total trajectories were selected". Section 10 in the Methods further details selection criteria, but is phrased quite unclearly. Does it suggest that only those trajectories were selected that showed all three behaviors? Or that only trajectories were selected that each showed only one of the behaviors? The fact that only 1% of the trajectories were selected for analysis raises a concern about them being representative. The authors will have to introduce analyses that shows that they are showing behavior that holds for the majority.

- The manuscript should be carefully proofread by a native English speaker. The text contains many terms that are awkward and each have quite sensible alternatives more stylistically appropriate. For example, in the abstract, "the genetic code carrier molecule", and "outstretched DNA".

- Reference 20 and 34: "Science (80-.)"?

- Reference 3: formatting issue in page numbers

Reviewer #3 (Remarks to the Author):

The work presented here from Ahmadi et al. details the characterization of EndoV at the single molecule level, and the topic of the manuscript: mechanisms of biological target searches and facilitated diffusion are of particular interest in the biophysics community. However, the current manuscript is lacking in biological context and could benefit from some consideration of the data analysis. As a result, we do not feel that the manuscript, " Breaking the speed limit: multimode fast scanning of DNA by Endonuclease V" is appropriate for publication in Nature Communications. Before resubmission, we encourage the authors to consider the following.

The present study characterizes the search mechanism of endoV, which the enzyme must use to locate damaged bases in vivo. Yet, in the current work there is no investigation of how the diffusion of the enzyme is modulated when encounters a template with a deaminated base. The author's model for endoV's activity would make some predictions about how the enzyme should interact with bona fide targets that would help to provide further support for the author's model, as well as provide more biological context to the data.

The author's model also makes some predictions about how the search mechanism of endoV should proceed in the presence of crowding on the substrate. To give some cellular context to the data, the differences between the proposed diffusion modes could be tested by using DNA templates containing bound roadblocks. Based on the proposed model, a congested construct would have predictable effects on the motion of the protein depending on which diffusion mode the enzyme is in (or has access to) when it encounters a roadblock.

The above are just suggestions, there are other potential experiments that could enrich the manuscript. Regardless of how, we feel that the study needs to build off the initial observations, and test them against some aspect of the enzymes cellular life.

Additionally, there are a few methods in the data analysis that seem potentially problematic. However, it may be that the reviewers have misunderstood the operations being performed upon the data, if so this misunderstanding highlights a need for revision of the methods with an eye toward additional clarity. The major points in the analysis that need revisiting are:

It is unclear whether the different populations of calculated diffusion coefficients represent a significant signal within the data when compared to those found via simulation. Furthermore, the data merits two additional diffusion models, one that switches in between a simple random walk

and a "confined" state, and another that has three states: confined, slow, and fast. These models should be analyzed side by side with the included model and data in the paper. Together with these data, a conversation is warranted in the methods that controls for sampling depth and how likely the observed distributions are from the data and from the models.

As a further point on the above, the use of machine learning algorithms to categorize segments of measured trajectories is a powerful method, but because of its complexity in application it is hard for the reader to quickly determine whether the settled on criteria used to make selections is robust and how changes in those criteria would affect the interpretation of results. A comparison of the categories identified by machine learning either against a spectrum of parameter space or against categorization via conventional methods would validate the use of the machine learning methods. The results of this characterization would merit a figure panel in the main text.

I review non-anonymously Sy Redding (UCSF). Furthermore this review was prepared with the assistance of Dr. Lucy Brennen (Redding Lab, UCSF). Additionally, I would like to suggest that the authors publically post their work on a preprint server (i.e. BioRxiv). This will give the authors input from the rest of the scientific world to help strengthen their manuscript to publication. As an additional point we would like to extend our appreciation of the inclusion of the data analysis code and workbooks that were made available in the supplement. These valuable resources too should be made available to the community through an online server (i.e. github).

Response: The authors would like to thank the referees for their thorough review of the original manuscript, and for the critical points which they raised. In answering these points, we feel that the quality of the manuscript has been raised, and appreciate the contribution of the referees to this improvement.

One topic was raised in a variety of overlapping forms by **all three referees** and it seems best to provide a common answer here, for the sake of coherence. The choice of a machine learning approach to classify segments of data into three speed classes according to heuristics determined from a hand-curated training set was criticized from multiple angles. Referee #1 argues that using a hidden Markov method (vbSPT) to identify the correct number of hidden states would be both more objective and a fundamentally better approach. Referee #2 brings up a hallmark paper which provides a theoretical activation energy-based description of the transition between “scanning while sliding and not scanning while hopping,” which can also be read as a superior alternative to the machine learning approach we have taken. Referee #3 requests a comparison between the fits of a two-state model and a three-state model, and a deeper understanding of the details of the machine learning method.

The primary function of the machine-learning classification was to allow salt-dependence to be investigated separately for qualitatively different classes of movement within trajectories, and to allow comparison of scanning behavior between different proteins using the same model, rather than to draw detailed conclusions about the individual classifications. Since all referees request or suggest the use of accepted methods for classification, we have chosen to replace the machine learning approach with the combined (and very concurrent) results of vbSPT and activation energy classifications.

These results satisfy the requirements of the referees, while making the machine-learning results redundant. Details related to specific requests related to this will be dealt with point by point in the response below.

Reviewers' comments:

Reviewer #1 (Remarks to the Author):

The authors perform particle tracking experiments on enzymes involved in DNA repair. The main conclusion is that Endonuclease V has at least three distinct motion states associated with DNA: confined, slow, and fast. All three are passive and thermally driven. The fast state is speculated to be diffusive electrostatic gliding along DNA, which is faster than the standard helical sliding seen in related enzymes. The characteristics and dynamics of the motion states are significant as they directly influence the rate of repair; i.e., motion is optimized for searching out damaged segments of DNA. The main conclusion is supported by analysis of particle tracks obtained from fluorescence microscopy. The authors use fluorophores that have a negligible impact on the hydrodynamic radius of observed enzymes. The conclusion is first supported by computing "instantaneous diffusion" histograms over all particle tracks, which amounts to convolutionally estimating effective diffusion in a sliding window. Different window sizes were shown not to affect the qualitative shape of the histograms. The diffusion histograms for WT-EndoV was shown to be bimodal, with two well separated (~10 fold) peaks. A control wm-EndoV and a simulated control (Brownian motion) were shown to have a single peak. A similar double peak was observed for a related enzyme hOGG1. This data supports at least two modes of motion. No third mode is observed corresponding to fast motion. The justification for the third mode is the relative fraction of observed instantaneous diffusivities above the theoretical maximum for helical sliding (2% for hOGG1 and 14% for WT-EndoV).

To gather further justification for the three mode hypothesis, the authors use a machine learning method to classify motion states over 5 increment windows of a subset of tracks. A smaller subset of tracks is divided from the rest and hand segmented; each frame is labeled as confined, slow, or fast. Parameters of a random forest model are trained with the hand segmented tracks and then deployed on the remainder of the data. The total fraction of time the enzymes spend in each of the three states is computed from the classified tracks. The fraction of time in the confined state increases whereas the fast state is preserved between the wt-EndoV and the mutant, suggesting the wedge motif is essential for helical sliding but not fast motion.

Clearly, it is an important problem to understand dynamics of stochastic motion of DNA repair enzymes, and the authors have collected extremely valuable data. However, I am not convinced of their conclusion of a distinct third motion state. Unless substantial revisions are made, I cannot recommend publication.

1. The instantaneous diffusion result would better support a third motion state if there were three distinct peaks. The existence of 14% outliers above a theoretical maximum diffusivity is weak evidence given the many sources of noise that can affect tracking measurements. Minimally, one could show that a three state Markov process, with parameter fits from data, reconstitutes the observed histogram, while a best fit two state process does not.

Response: In the revised version of the paper, and as explained at the beginning, we have taken the suggestions of referees #1 and #2 on board. Referee #1 brings up the three state Markov process in point 2 below, so this request is answered separately. We have now also placed our results in the context of the hallmark paper by Slutsky and Mirny (2004; *Biophys. J.*, 87, 4021-35)¹ referred to by referee #2. We segmented the data using the concept of activation energy barriers and show that this is compatible with there being three distinct modes of scanning based on three different ranges of activation barriers (lines 143-169). The occupancy of the high-speed state is in the range of 30-45 % for both the hidden Markov and energy barrier segmentation methods for EndoV.

As is stated in the revised manuscript (lines 222-225), it has previously been shown that for typical DNA proteins friction reduces the diffusion rate by a factor of 2-5 compared to the theoretically calculated upper limit of diffusion². Therefore scanning with a diffusion rate close to and greater than the theoretical limit is significant. In addition, the histogram is the aggregated analysis of over 3000 trajectories and since the horizontal axis in instantaneous diffusion rate is on a logarithmic scale, values exceeding the speed limit are significant and frequent making it unlikely to be noise or outliers. Moreover the existence of the third mode of scanning is entirely compatible with a 1D hopping mode, which – in agreement with accepted theory – shows a salt-concentration dependence. This has been verified by the revised analysis (lines 222-246 and Fig.3)

In addition we have applied the hidden Markov analysis with 3 states on our data – see answer to point 2. The result is presented in Figure 2b in the revised manuscript and shows very good agreement between the two independent classification methods. This agreement was not present when we ran the hidden Markov analysis with 2 states.

I am aware the authors did experiments adding salt, but it is outside my expertise to say if this is significant evidence to support the main conclusion by itself, the tracking analysis certainly does not.

Response: We find that the two slower modes show no dependence on salt concentration - an observation that implies constant contact between the enzyme and the DNA. Figure 3 clearly shows a different behavior for the fast state, where the diffusion rate rises with increasing salt-concentration, which implies that the salt screens/shields the DNA and increases the chance of microscopic dissociation and spending more time in hopping. Such salt-dependence of diffusion rate is well accepted in the literature³⁻⁵ and allows us to infer the third distinct mode. We have covered this in the revised manuscript (lines 233-243)

2. The machine learning approach (i.e., hand segmentation of tracks) inserts human bias into the classification scheme. Three motion states were classified because the model was trained by hand segmented tracks. At least the same bias is uniformly applied to all of the data. Sometimes this is the best that can be done. However, in this case, much better methods are readily available. Namely, the EM algorithm for the hidden Markov models approach, which is a systematic Bayesian formalism that has numerous advantages over hand segmentation: no hand segmentation is necessary, all of the assumptions are placed into a physical model (Markov process) of particle motion, and variational Bayes methods can even quantitatively select the best model (there is a good Matlab package freely available from the Elf lab <http://vbspt.sourceforge.net/> <<http://vbspt.sourceforge.net/>>). And I think these methods should be used to make the current work more objective and to encourage others to use the best methods available.

Response: As explained previously, since we have performed a segmentation based on activation energy barriers, there is no need for hand segmentation and machine learning analysis. Nevertheless, we verified this energy barrier-based classification using the hidden Markov model as suggested. We used the recommended Markov analysis software (<http://vbspt.sourceforge.net/>) on our data. In the first step, to determine the ideal number of states by overspecifying the initial number of hidden states, vbSPT consistently selected more than three states to be best fit for all proteins (8 states for wt-EndoV), while it reliably selected a single mode as the best fit for the simulated random walk. In discussion with the authors of the code, we concluded that this is caused either by heterogeneity in the trajectories or by overfitting due to the diffusion model that the code uses in which localization error or motion blur are neglected. Irrespective of this, the clear conclusion is that vbSPT prefers a 3 or more state model to a 2 state model.

Following the suggestion of the lead author of vbSPT we set the number of hidden states to 3 and compared the result of segmentation with the energy barrier segmentation. We find a striking agreement between the state occupancy and diffusion rate of the two models (Fig. 2b). In addition these two models show a strong overlap of classification (with accuracy of 73.4%) as shown in the confusion matrix here:

	Energy barrier		
Markov model	1	2	3
1	28643	8107	835
2	2089	18523	9599
3	487	4435	23346

	Class: 1	Class: 2	Class: 3
Sensitivity	0.9175	0.5963	0.6911
Specificity	0.8621	0.8202	0.9210

Reviewer #2 (Remarks to the Author):

The authors use single-molecule fluorescence imaging of a labeled DNA-binding protein (a bacterial endonuclease) on stretched DNA to visualize the molecular mechanisms with which the protein moves one-dimensionally along DNA. By an improved temporal resolution, they show that the protein transitions between two binding modes of 1d translocation: one that corresponds to helical scanning, the other to a nonhelical, microscopic hopping. While the observation is interesting and provides another example of how DNA-binding proteins have evolved to combine several search mechanisms to achieve overall kinetic efficiency in target identification, the work is largely confirming previously observed behavior of other systems.

In particular, similar work on the PCNA sliding clamp (Ref 17) demonstrated an alternation between helical and non-helical scanning, and other work on a DNA-bound phage polymerase (PMID: 20080681) showed that a similar transition could be induced by addition of a co-factor. The authors argue in their introduction that the behavior they observe with EndoV is different: Lines 57-59, “this has so far only been attributed either to...or to conformational changes” an argument repeated in their conclusion (lines 160-161). This reasoning is incorrect. Whatever the transitioning between the helical scanning mode and the 1d hopping mechanism, it is caused by a conformational change. It might be changing the tilt angle of a protein on the DNA (PCNA), binding/dissociation of a protein factor (the phage polymerase) or insertion/removal of a wedge motive (this work), it is still a conformational change. It might be thermally driven (line 63), it is still a conformational change...

Response: We appreciate the reviewer’s comment and we have clarified this issue in the revised manuscript (line 54-60, 96-98, 259-269). We feel that there is something of a misunderstanding here. While conformational states of the complete protein-DNA complex changes when switching between hopping and sliding - including the angle at which the protein approaches the DNA, electrostatic interactions etc - we exclusively mean changes in the conformation of the protein, not the complex, when we write “proteins which undergo cofactor-induced conformational changes”.

Our data show that the wedge motif of EndoV has no detectable impact on switching between helical sliding and hopping modes since the hopping behavior is consistent for both wt- and wm-EndoV (Fig. 2a and Fig. 3). The wedge motif does however affect switching between interrogation (confined) and helical sliding (slow) modes. Further, crystallographic data shows that both free and DNA-bound monomeric EndoV are structurally similar⁶. In our opinion, these observations rule out structural changes *of the protein* as the mechanism of mode switching between hopping and helical sliding.

Both of the enzymes studied here are monomeric globular proteins whose interaction with DNA differs from the other systems described in the literature. For example, PCNA has a closed topological shape around DNA that keeps the multimeric protein in the vicinity of DNA at all times; this plays an important role in facilitating hopping⁵. MutS α ⁷ and EcoP15I⁸ change protein conformation with ATP association/hydrolysis, and p53 has two different binding sites and each mode of scanning is delegated to one of these binding sites³. TALE uses two distinct protein conformations in either scanning or recognition mode, while in the case of T7 DNA polymerase⁹ the change of the scanning mode is linked to association of a cofactor.

We argue that the switching between helical sliding and hopping, as observed here for EndoV, therefore points to the possibility that switching between hopping and helical sliding is not only associated with

proteins that undergo ATP-induced protein conformational changes, encircle DNA or display several DNA binding sites, but can be a more general strategy of DNA scanning adopted by some, though not necessarily all, smaller DNA scanning proteins.

The authors also ignore a hallmark paper in this field that provided more than a decade ago the theoretical rationalization for this thermally induced transition between scanning while sliding and not scanning while hopping (PMID: 15465864). Later experimental work (including the examples above) was interpreted in the context of this thermally-induced two-mode sliding model and has resulted in a picture of target identification on DNA that is now commonly accepted in the field. While interesting in its high time resolution and the direct observation of these two distinct diffusional behaviors, the work does not provide a significant advance in our understanding of the underlying mechanisms. The experiments are sound and the interpretation supported by the data. As such, my recommendation would be to consider a more specialized journal.

Response: As mentioned at the beginning in our reply to all referees, we have studied this paper¹ (PMID: 15465864) alongside other experimental papers that use the same rationalization to explain scanning mechanisms^{3,5,10-12} and found it to be very relevant. In the revised version of the manuscript, the trajectories are classified into three scanning modes based on three different ranges of activation energy barriers adopted from the mentioned studies (lines 61-76, 143-169). We have interpreted our results in the light of this analysis, combined with the hidden Markov results, which are in good agreement.

- Line 43-44, “with a 100-fold increased rate”. The authors should be careful with absolute statements such as this one. While 1d scanning can speed up target association, the rate with which this occurs is entirely dependent on the exact parameters, in particular the balancing between the association equilibrium constant of the protein with the DNA and the scanning rate constant, and the division between helical scanning and 1d hopping. A very tight association with the DNA while 1d scanning might even result in slower searches than exclusively 3d! See for example PMID: 15465864. This latter paper also lays the theoretical basis for the key observation in the current manuscript: the conformational transitioning between a mode that corresponds with sequence sampling and a mode that corresponds to mere 1d movement (without scanning). The authors should discuss this paper and place their results in the context of this theoretical framework.

Response: We have modified the “with a 100-fold increased rate” to “with a dramatically increased rate” (line 39).

We used the concept of energy barriers of scanning from the mentioned study to address the different scanning modes. We have also discussed the role of the wedge motif in switching between different scanning modes, by comparing the proportion of time that wt-EndoV and wm-EndoV spend in interrogation mode (activation energy $> 2k_B T$) (lines 196-221).

Moreover using the proposed values of scanning rate constant for helical sliding in the theory paper¹ and previous experimental reports^{3,5,10-12}, we distinguished between the helical sliding and hopping (lines 143-169) and further verify that hopping is a fundamentally different interaction with the DNA by showing that it has a salt-dependent diffusion rate (line 222-246 and Fig. 3).

- Lines 89-91, “the payload of large antibodies and quantum dots...used...in previous studies...have been replaced...”. This is quite a disingenuous statement. A large part of the reported work in this area has been done using small-molecule organic dye molecules.

Response: This clumsy wording has been corrected (lines 111-117). The point was to stress that data was collected using small-molecule dyes and in conditions with no external flow or need for staining of DNA, but using a laser tweezers to localize and stretch the DNA, a combination that reduces errors in data collection.

- Line 120, “Around 1% of the total trajectories were selected”. Section 10 in the Methods further details selection criteria, but is phrased quite unclearly. Does it suggest that only those trajectories were selected that showed all three behaviors? Or that only trajectories were selected that each showed only one of the behaviors? The fact that only 1% of the trajectories were selected for analysis raises a concern about them being representative. The authors will have to introduce analyses that shows that they are showing behavior that holds for the majority.

Response: As it is mentioned in the response to the previous comments, in the revised version of the manuscript, the trajectories are classified based on the activation energy barriers of DNA scanning as outlined in the Slutsky and Mirny 2004 paper (lines 143-169) and this classification is shown to be coherent with the Markov hidden state analysis as reviewer 1 suggested (lines 170-188).

The 1% referred to a hand-selected sub-set used for training the machine-learning model initially, while all of the trajectories were subsequently analyzed. This analysis is no longer part of the manuscript as it was replaced by the new classification methods.

- The manuscript should be carefully proofread by a native English speaker. The text contains many terms that are awkward and each have quite sensible alternatives more stylistically appropriate. For example, in the abstract, “the genetic code carrier molecule”, and “outstretched DNA”.

Response: This has been corrected.

- Reference 20 and 34: “Science (80-.)”?

Response: This has been corrected.

- Reference 3: formatting issue in page numbers

Response: This has been corrected.

Reviewer #3 (Remarks to the Author):

The work presented here from Ahmadi et al. details the characterization of EndoV at the single molecule level, and the topic of the manuscript: mechanisms of biological target searches and facilitated diffusion are of particular interest in the biophysics community. However, the current manuscript is lacking in biological context and could benefit from some consideration of the data analysis. As a result, we do not feel that the manuscript, “ Breaking the speed limit: multimode fast scanning of DNA by Endonuclease V”

is appropriate for publication in Nature Communications. Before resubmission, we encourage the authors to consider the following.

The present study characterizes the search mechanism of endoV, which the enzyme must use to locate damaged bases in vivo. Yet, in the current work there is no investigation of how the diffusion of the enzyme is modulated when encounters a template with a deaminated base. The author's model for endoV's activity would make some predictions about how the enzyme should interact with bona fide targets that would help to provide further support for the author's model, as well as provide more biological context to the data.

Response: The reviewer brings up an interesting issue – how does EndoV behave when encountering a deaminated base, to look at the switching between the stationary and helical scanning modes. However, we think the suggestion of including DNA scanning experiments with a DNA template containing a damaged nucleotide is so challenging that it merits a separate study. First, preparing a DNA with a deaminated base in a specific position in a single DNA molecule, and without other damages present that could interfere with the analysis, is going to be challenging. This notion is supported by the fact that biochemical data shows that EndoV also detects and cleaves DNA other than at deaminated adenines, such as base mismatches, insertion-deletion loops, and probably most importantly abasic sites, which are all characterized by forming an inherent weak point in the DNA stack. The presence of any such element in the DNA would interfere with data analysis. Second, it is known that EndoV binds strongly to the cleaved DNA product, hence it would be unexpected to see much DNA scanning after damage detection/cleavage. The molecular basis for this incision has been described in detail by our lab in the past¹³ (Dalhus et al, Nature Struct Mol Biol, 2009). The enzyme is probably protecting the cleaved DNA (which is very toxic to any cell) until the next enzyme in the repair pathway takes over. The pathway for EndoV is still not known, so we are not able to add that second factor into the experiment either. We have plans to carry out such two-component, DNA hand-over experiments for other DNA repair pathways, but more work is needed with respect to DNA preparation and flow-cell design.

With respect to the biological role of EndoV, we have included more biological data in the manuscript and placed our findings in a biological setting (144-169). One of the key findings is that the EndoV wedge motif is affecting the switching between interrogation (confined) and helical sliding (slow) modes, where the wedge probably plays an important role in separating the DNA strands close to the lesion (202-217).

The author's model also makes some predictions about how the search mechanism of endoV should proceed in the presence of crowding on the substrate. To give some cellular context to the data, the differences between the proposed diffusion modes could be tested by using DNA templates containing bound roadblocks. Based on the proposed model, a congested construct would have predictable effects on the motion of the protein depending on which diffusion mode the enzyme is in (or has access to) when it encounters a roadblock.

Response: We agree with the reviewer that it would be very interesting to investigate DNA templates with roadblocks, and observe possible hopping across such barriers. Biologically relevant steric roadblocks include insertion-deletion loops (known as mismatch loops) and 3'- or 5'-flaps, which are all substrates for EndoV. However, as with the deaminated substrate described above, it will be challenging to implement this type of substrates for single molecule experiments and is therefore planned as a subsequent study. Another interesting DNA block would be nucleosome core particles, however, only bacterial EndoV has DNA as substrate – the mammalian versions of EndoV cleave RNA (Vik et al, Nature

Comm. 2013), which does not contain nucleosomes. The main focus of the present work has thus been to identify and characterize various scanning modes of EndoV on non-damaged DNA.

The above are just suggestions, there are other potential experiments that could enrich the manuscript. Regardless of how, we feel that the study needs to build off the initial observations, and test them against some aspect of the enzymes cellular life.

Response: Previously, we and others have characterized the biochemical and structural properties of EndoV extensively, including enzymology and DNA substrate recognition. In the revised version we are discussing our single molecule data presented in this work in view of our current knowledge about EndoV.

Additionally, there are a few methods in the data analysis that seem potentially problematic. However, it may be that the reviewers have misunderstood the operations being performed upon the data, if so this misunderstanding highlights a need for revision of the methods with an eye toward additional clarity. The major points in the analysis that need revisiting are:

It is unclear whether the different populations of calculated diffusion coefficients represent a significant signal within the data when compared to those found via simulation. , the data merits two additional diffusion models, one that switches in between a simple random walk and a “confined” state, and another that has three states: confined, slow, and fast. These models should be analyzed side by side with the included model and data in the paper. Together with these data, a conversation is warranted in the methods that controls for sampling depth and how likely the observed distributions are from the data and from the models.

Response: As covered in the answer to all authors at the beginning, and fully discussed in the replies to referees #1 and #2, this question has been answered using the two acceptably rigorous methods suggested by the referees. The conclusion of both methods is that a three state model is the most suitable representation of the data (lines 143-188 and Fig. 2).

In order to guide the eye, we have also simulated monomodal random walks with the average diffusion rate corresponding to that measured for each of the three proteins. In figure 2a of the revised version of the manuscript, we have plotted the measured distributions for each protein with the corresponding simulated random walk distribution overlaid. This nicely illustrates how different the distribution of multi-mode diffusion rates is from a representative monomodal random walk. The three modes of diffusion that are grouped according to the activation energy barrier of sliding are also colored in the histogram, showing how these groups also relate to a straightforward simulated random walk.

As a further point on the above, the use of machine learning algorithms to categorize segments of measured trajectories is a powerful method, but because of its complexity in application it is hard for the reader to quickly determine whether the settled on criteria used to make selections is robust and how changes in those criteria would affect the interpretation of results. A comparison of the categories identified by machine learning either against a spectrum of parameter space or against categorization via conventional methods would validate the use of the machine learning methods. The results of this characterization would merit a figure panel in the main text.

Response: As previously stated, we have chosen to interpret our results based on the following theory paper¹ (PMID: 15465864) and the vbSPT (<http://vbspt.sourceforge.net/>) software from the Elf lab. As a result, the machine learning classification is no longer included in the manuscript.

I review non-anonymously Sy Redding (UCSF). Furthermore this review was prepared with the assistance of Dr. Lucy Brennen (Redding Lab, UCSF). Additionally, I would like to suggest that the authors publically post their work on a preprint server (i.e. BioRxiv). This will give the authors input from the rest of the scientific world to help strengthen their manuscript to publication. As an additional point we would like to extend our appreciation of the inclusion of the data analysis code and workbooks that were made available in the supplement. These valuable resources too should be made available to the community through an online server (i.e. github).

Response: We appreciate your choice to review non-anonymously. We believe it is a worthwhile contribution to the long term integrity of the publication process.

All of the source code and data is available on github and will be made public once the manuscript is published. We fully intend to use bioRxiv in the future. A certain lack of clarity over how journals respond to preprints and copyrights has discouraged us previously, but times are changing.

Unfortunately the use of the vbSPT software suggested by Referee #1 requires the use of Matlab, and therefore breaks our model of a completely open reproducible pipeline for data analysis, but without rewriting the entire method in an open language, we have no choice but to include this approach. We hope that newer versions of the vbSPT software will be written in a less isolated form, so that they can be included in truly open research protocols.

1. Slutsky, M. & Mirny, L. A. Kinetics of Protein-DNA Interaction: Facilitated Target Location in Sequence-Dependent Potential. *Biophys. J.* **87**, 4021–4035 (2004).
2. Bagchi, B., Blainey, P. C. & Sunney Xie, X. Diffusion constant of a nonspecifically bound protein undergoing curvilinear motion along DNA. *J. Phys. Chem. B* **112**, 6282–6284 (2008).
3. Tafvizi, A., Huang, F., Fersht, A. R., Mirny, L. a & van Oijen, A. M. A single-molecule characterization of p53 search on DNA. *Proc. Natl. Acad. Sci. U. S. A.* **108**, 563–8 (2011).
4. Komazin-Meredith, G., Mirchev, R., Golan, D. E., van Oijen, A. M. & Coen, D. M. Hopping of a processivity factor on DNA revealed by single-molecule assays of diffusion. *Proc. Natl. Acad. Sci.* **105**, 10721–10726 (2008).
5. Kochaniak, A. B. *et al.* Proliferating cell nuclear antigen uses two distinct modes to move along DNA. *J. Biol. Chem.* **284**, 17700–17710 (2009).
6. Dalhus, B. *et al.* Structures of endonuclease V with DNA reveal initiation of deaminated adenine repair. *Nat. Struct. Mol. Biol.* **16**, 138–143 (2009).
7. Gorman, J. *et al.* Single-molecule imaging reveals target-search mechanisms during DNA mismatch repair. *Proc. Natl. Acad. Sci. USA* **109**, E3074–E3083 (2012).
8. Schwarz, F. W. *et al.* The Helicase-Like Domains of Type III Restriction Enzymes Trigger Long-

Range Diffusion Along DNA. *Science* (80-.). **340**, 353–356 (2013).

9. Etson, C. M., Hamdan, S. M., Richardson, C. C. & van Oijen, A. M. Thioredoxin suppresses microscopic hopping of T7 DNA polymerase on duplex DNA. *Proc. Natl. Acad. Sci.* **107**, 1900–1905 (2010).
10. Blainey, P. C., van Oijen, A. M., Banerjee, A., Verdine, G. L. & Xie, X. S. A base-excision DNA-repair protein finds intrahelical lesion bases by fast sliding in contact with DNA. *Proc. Natl. Acad. Sci. U. S. A.* **103**, 5752–5757 (2006).
11. Tafvizi, A. *et al.* Tumor suppressor p53 slides on DNA with low friction and high stability. *Biophys. J.* **95**, L01–L03 (2008).
12. Dunn, A. R., Kad, N. M., Nelson, S. R., Warshaw, D. M. & Wallace, S. S. Single Qdot-labeled glycosylase molecules use a wedge amino acid to probe for lesions while scanning along DNA. *Nucleic Acids Res.* **39**, 7487–7498 (2011).
13. Dalhus, B. *et al.* Structures of endonuclease V with DNA reveal initiation of deaminated adenine repair. *Nat. Struct. Mol. Biol.* **16**, 138–143 (2009).

Reviewers' comments:

Reviewer #1 (Remarks to the Author):

The authors made substantial improvements. I particularly like the energy barrier perspective and the match between the two 3-state models. I think that they are close to a strong paper. I have a few more comments concerning the new HMM-based analysis.

1. What precisely is the model used for HMM? It should be a simple Gaussian mixture: a set of states, each with its own diffusion coefficient, along with transition rates, all to all, for jumping between each state. It should have diffusion coefficients for each state. The output of vbSPT should then include a maximum likelihood (ML) fit. Did the authors use a different estimation method for the diffusivities they reported? If so, why? ML fits should also be available for the transition rates. The inferred transition rate matrix can give valuable information about the dynamics of the process. In particular, it may shed some light on why the vbSPT method converged to more than three states.

2. "... the software failed to converge on a reasonable number of states, and always selected more than three states to be the best fit for all proteins (eight states for wt-EndoV) with negligible occupancy for some of the states."

Just because state occupations are negligible, does not mean they are not important or meaningful. Enzyme complexes are often short lived, but clearly important. I can think of two reasons additional states might be relevant. (i) There are non-Markovian transitions (memory) between the three sliding states, and vbSPT is using additional Markovian states to approximate this. In this case, it is the dynamics of switching that must be further investigated. (ii) there is population heterogeneity (different types), and vbSPT is trying to capture that by adding multiple copies of the same three states so that individual observed tracks are either in one set of states or the another. This should show up in the inferred transition rate matrix as a (nearly) reducible matrix. In other words, it is effectively impossible to reach any state, through a sequence of transitions, given any starting state.

3. "We concluded that this is caused either by heterogeneity in the trajectories or by overfitting due to the diffusion model that the code uses, in which localization error or motion blur are neglected."

This is a low-effort dismissal of an inconvenient outcome and needs more attention. The first possibility seems plausible and can be explored as described above. Based on the amount of data the authors are using, it seems very unlikely that the problem is overfitting. Localization error and motion blur can cause anomalous correlations between increments, which can show up in a micro rheology analysis when beads are in a highly elastic complex fluid, but this should be largely ignored by a Gaussian mixture model.

4. The poor match between the two state models compared to the 3-state models is a nice result! It should be emphasized more in the main text.

Reviewer #2 (Remarks to the Author):

The authors have responded to all the points raised by the reviewers and have produced a significantly improved manuscript. The work seems ready for publication and will be a nice addition to the field.

Response: The authors would once again like to thank reviewer #1 for insightful additional comments that, after further investigation of scanning mode transitions, we feel have raised the quality of the manuscript.

Reviewer #1 (Remarks to the Author):

The authors made substantial improvements. I particularly like the energy barrier perspective and the match between the two 3-state models. I think that they are close to a strong paper. I have a few more comments concerning the new HMM-based analysis.

1. What precisely is the model used for HMM? It should be a simple Gaussian mixture: a set of states, each with its own diffusion coefficient, along with transition rates, all to all, for jumping between each state. It should have diffusion coefficients for each state. The output of vbSPT should then include a maximum likelihood (ML) fit. Did the authors use a different estimation method for the diffusivities they reported? If so, why? ML fits should also be available for the transition rates. The inferred transition rate matrix can give valuable information about the dynamics of the process. In particular, it may shed some light on why the vbSPT method converged to more than three states.

Response:

(1) The vbSPT software represents the total movement as a mix of simple Gaussian functions, each with a diffusion coefficient. In such a model, it is assumed that the protein has different states of movements between which they perform memoryless transitions. Using a maximum-evidence approach, the parameters of the states, i.e. the diffusion coefficient, transition probabilities and the ideal number of states, are estimated for the best fit of the experimental data using a variational Bayesian treatment of the HMM. **We have included a section describing the model in the revised manuscript (lines 170-176).**

(2) Yes, we recalculated the diffusivity of each Markov state after segmentation in order to make sure we compare the exact same set of frames as used in the calculation of diffusivity for the energy barrier classification. In the energy barrier model, the frames are classified based on the value of the observed instantaneous diffusion rate. Since we use a moving window of 5 frames to calculate the instantaneous diffusion rate, the last four frames of each trajectory are not assigned any value for the instantaneous diffusion rate, and are thus excluded from the energy barrier classification. Because of this, we excluded the same four frames of each trajectory after the HMM classification, and recalculated the diffusion rate of each state by averaging over the instantaneous diffusion rate of all remaining frames. The occupancy of each state was also recalculated as the proportion of time spent in each state. The recalculated values are very close to the original values reported by the HMM without frame correction (Figure 1 below). **We have clarified this in the revised version of the manuscript (lines 200-206), and included a figure (Supplementary information, paragraph 7.2) showing the corrected and uncorrected parameters from the HMM classification along with the corresponding values from the energy barrier classification.**

Figure 1. Comparison of recalculated values for the diffusion rate and occupancy of the HMM states with the original output of vbSPT along with the corresponding values from the energy barrier classification. This figure has been included in the Supplementary Info, section 7.2.

(3) As suggested by the reviewer, we looked into the transition rates of the HMM states. This gave valuable insight into the mechanism of scanning of these proteins, and it also gave clues to the overestimation of the number of states by the software (explained in the response to the next questions). **We have included the transition probabilities for the 3-state models as a separate panel in Figure 2 in the revised manuscript, together with a brief discussion of the mechanistic interpretation of these transition probabilities (lines 211-217).**

2. "... the software failed to converge on a reasonable number of states, and always selected more than three states to be the best fit for all proteins (eight states for wt-EndoV) with negligible occupancy for some of the states."

Just because state occupations are negligible, does not mean they are not important or meaningful. Enzyme complexes are often short lived, but clearly important. I can think of two reasons additional states might be relevant. (i) There are non-Markovian transitions (memory) between the three sliding states, and vbSPT is using additional Markovian states to approximate this. In this case, it is the dynamics of switching that must be further investigated. (ii) there is population heterogeneity (different types), and vbSPT is trying to capture that by adding multiple copies of the same three states so that individual observed tracks are either in one set of states or the another. This should show up in the inferred transition rate matrix as a (nearly) reducible matrix. In other words, it is effectively impossible to reach any state, through a sequence of transitions, given any starting state.

Response: We appreciate the insight that the reviewer provided us on this issue. To investigate the possible existence of non-Markovian transitions, we checked the dynamics of switching between scanning modes for all combinations of three consecutive states for all proteins. There are 12 alternative sequences of three consecutive states. We calculated the relative probabilities for the two possible outcomes of the second transition, given the same initial transition. The combined result of this analysis is shown in Figure 2 below. The x-axis shows the initial transition and the y-axis is the relative probability of transition to either of two possible states given the middle state.

For wt-EndoV (Figure 2 below, middle panel) we see that for the transition from 1 to 2 ($1 > 2$), the chance of going back to state 1 is around twice (~68%) that of going to state 3 (~32%). For the transition from 3 to 2 ($3 > 2$), the situation is reversed, and it is more likely that protein goes back to state 3 (~62%)

than going to state 1 (~38%). In other words, the transition for leaving state 2 (helical sliding) is dependent on the transition going into state 2 in such a way that the protein molecules prefer to go back to the same state as before. The situation is the same for the wedge-mutant EndoV (Figure 2 below, right panel). Thus, we interpret this as “memory” between transitions for sequences of transitions in which the middle state is 2 (helical sliding).

The presence of such non-Markovian transitions in these sequences of transitions can, as pointed out by the reviewer, explain the tendency for vbSPT to overestimate the number of diffusion states. **We have revised the manuscript and included a brief discussion of the major findings of this analysis (lines 180-188), as well as included the transition probabilities shown in Figure 2 below as a figure in the Supplementary info, paragraph 7.2.**

Figure 2. Detection of non-Markovian transitions. Analysis of all combinations of two consecutive transitions for all three proteins.

3. "We concluded that this is caused either by heterogeneity in the trajectories or by overfitting due to the diffusion model that the code uses, in which localization error or motion blur are neglected."

This is a low-effort dismissal of an inconvenient outcome and needs more attention. The first possibility seems plausible and can be explored as described above. Based on the amount of data the authors are using, it seems very unlikely that the problem is overfitting. Localization error and motion blur can cause anomalous correlations between increments, which can show up in a micro rheology analysis when beads are in a highly elastic complex fluid, but this should be largely ignored by a Gaussian mixture model.

Response: In addition to the determination of the presence of a certain proportion of non-Markovian transitions in the data, as discussed above, we investigated the possible role of population heterogeneity in overestimation of the number of HMM states. We checked the transition probabilities of the Markov chains for all models with more than 3 states. For all of these models (13 in total; transition matrices are presented in Supplementary Info 7.2), we find that 11 are reducible Markov chains (similar to example shown below for 5-state of wt-EndoV). For the remaining 2 models (the 4-state models of wt- and wm-EndoV), the Markov chains are nearly reducible (example shown below for 4-state of wt-EndoV). In contrast, all the 3-state models presented in our study have irreducible Markov chains. From this inspection, we conclude that there are signs of heterogeneity in our data sets, which might further explain why the vbSPT HM modelling tends to find more than 3 states for our data sets when given the opportunity to do so. In this analysis of the transition matrices, we assigned all transition probabilities < 0.012 as being negligible, as it has been shown that the vbSPT software can report transition probabilities up to 0.12 even for forbidden transitions in a simulated data set¹. **We have listed all matrices for all predicted HMMs with more than 3 states in the Supplementary Info, paragraph 7.2, and a statement describing this has been added to the revised version of the manuscript (lines: 190-194).**

wt-EndoV 3-state

	1	2	3
1	0.957	0.029	0.014
2	0.032	0.931	0.037
3	0.014	0.053	0.934

wt-EndoV 4-state

	1	2	3	4
1	0.937	0.039	0.000	0.023
2	0.050	0.939	0.000	0.000
3	0.000	0.000	0.975	0.019
4	0.038	0.046	0.088	0.828

wt-EndoV 5-state

	1	2	3	4	5
1	0.954	0.000	0.037	0.000	0.000
2	0.000	0.777	0.017	0.000	0.206
3	0.047	0.000	0.944	0.000	0.000
4	0.000	0.000	0.000	0.984	0.000
5	0.021	0.051	0.038	0.084	0.807

4. The poor match between the two state models compared to the 3-state models is a nice result! It should be emphasized more in the main text.

Response: We have added a brief comment on the poor match for the 2-state models compared to the 3-state models in the revised version of the manuscript (lines: 194-196).

We have also added a few references (18, 50) and corrected the reference numbering in this revision.

1. Persson, F., Lindén, M., Unoson, C. & Elf, J. Extracting intracellular diffusive states and transition rates from single-molecule tracking data. *Nat. Methods* **10**, 265–9 (2013).

REVIEWERS' COMMENTS:

Reviewer #1 (Remarks to the Author):

The authors have fully addressed all of my comments. Once again, the authors have substantially improved the paper, beyond my expectations. I strongly recommend publication without further comment.